# Geographical Distribution of Periodontitis Risk and Prevalence in Portugal Using Multivariable Data Mining and Modeling

**DOI:** 10.3390/ijerph192013634

**Published:** 2022-10-20

**Authors:** Ana Antunes, João Botelho, José João Mendes, Ana Sintra Delgado, Vanessa Machado, Luís Proença

**Affiliations:** 1Clinical Research Unit (CRU), Centro de Investigação Interdisciplinar Egas Moniz (CiiEM), Egas Moniz—Cooperativa de Ensino Superior, Caparica, 2829-511 Almada, Portugal; 2Evidence-Based Hub, CiiEM, Egas Moniz—Cooperativa de Ensino Superior, Caparica, 2829-511 Almada, Portugal

**Keywords:** periodontitis, periodontal disease, prediction, prevalence, risk, modeling, public health, oral health

## Abstract

We aimed to estimate the geographical distribution of periodontitis prevalence and risk based on sociodemographic and economic data. This study used sociodemographic, economic, and health services data obtained from a regional survey and governmental open data sources. Information was gathered for all 308 Portuguese municipalities and compiled in a large set of 52 variables. We employed principal component analysis (PCA), factor analysis (FA) and clustering techniques to model the Portuguese nationwide geographical distribution of the disease. Estimation of periodontitis risk for each municipality was achieved by calculation of a normalized score, obtained as an adjusted linear combination of six independent factors that were extracted through PCA/FA. The municipalities were also classified according to a quartile-based risk grade in each cluster. Additionally, linear regression was used to estimate the periodontitis prevalence within the peri-urban municipality clusters, accounting for 30.5% of the Portuguese population. A total of nine municipality clusters were obtained with the following characteristics: mainly rural/low populated, including small villages (one), partly rural, including small cities (two), mainly urban/peri-urban, including medium-sized to large cities (4), and urban/large cities (2). Within the clusters, a higher periodontitis risk was identified for municipalities with lower income, older populations. The estimated periodontitis prevalence for the 18 municipalities included in the four peri-urban clusters ranged from 41.2% to 69.0%. Periodontitis prevalence estimates range from 41.2% to 69.0% for the municipalities characterized as peri-urban and mainly urban, most of them located in the Lisbon Metropolitan Area, the tenth largest in Europe.

## 1. Introduction

Periodontitis is a disease caused by a dysbiotic polymicrobial community surrounding teeth [1,2,3]. Without adequate treatment, it can lead to progressive periodontal destruction, tooth loss [4] decrease in oral health-related quality of life [5,6], and have an impactful economic burden [7]. Between 1990 and 2010, periodontitis was ranked one of the most prevalent noncommunicable diseases in adults [8,9], and a recent update confirmed that prevalence and incidence is increasing among younger individuals [10].

To date, the epidemiology of periodontal diseases in the Portuguese population has been inadequately reported. A single national epidemiological study, carried out in 2015, estimated a prevalence of 10.8% and 15.3% for the adult and elderly population, respectively [11]. These results contrasted with a regional prevalence of 59.9% reported in the study of periodontal health in Almada-Seixal (SoPHiAS), conducted in the southern Lisbon Metropolitan Area [12] and with European estimations that ranged between 38.4% and 89.7% [13,14,15,16,17,18]. Periodontitis risk and progression is known to be strongly dependent on several key indicators, including age, education, oral care access, deleterious behaviors (smoking and alcohol habits), socioeconomic status, and uncontrolled systemic diseases (i.e., diabetes mellitus and hypertension, among others) [19,20,21,22].

The current gap of knowledge on periodontal status among the Portuguese population highlights the need for further studies. In our view, a comprehensive projection of risk trends for the disease at both national and regional levels based on current evidence is relevant. The geographical distribution of periodontitis prevalence and risk estimates are critical for understanding the burden of this disease for each specific region. This permits conducting a region-targeted adequate planning of the oral health policy in order to prevent and give early access to treatment for those in need. In this study, we aimed to develop a model able to estimate periodontitis risk for the Portuguese population at regional/territory level based on a multivariate approach using governmental sociodemographic, economic, and health services data along with epidemiological data.

## 2. Materials and Methods

### 2.1. Study Area

Secondary data were included to characterize and fingerprint the sociodemographic, economic, and health profiles at municipality level. Portugal is divided into 308 municipalities, also known as local administrative units, level 1 (LAU 1).

The risk of periodontitis was considered the main outcome. It was estimated for each municipality by calculation of a numerical score, obtained as an adjusted linear combination of six independent factors that were extracted through a dimension reduction process. The normalized score, ranging from 0 to 25, was further used to categorize, within each cluster, the municipalities in a 4-level risk grade, identified by its quartile position (from Q1—higher risk to Q4—lower risk).

### 2.2. Variables

A large set of 52 variables was considered and the correspondent values gathered for each municipality.

The variables included data on the following areas: population, including sex, age and area distribution (population density); education; purchasing power; unemployment/employment; inactive/active population; longevity; aging; elderly dependence and health personnel (number of doctors, dentists, pharmacists and nurses).

All data were gathered from open public sources (official government statistics). Sociodemographic data were obtained from PORDATA, an official and certified statistics data website (https://www.pordata.pt, accessed on 1 May 2022). Overall, data included comprised age, sex distribution, male/female ratio, population density from the 2021 CENSUS, number of doctors, dentist, nurses and pharmacists in 2019, illiterates in total, education (categorized as no education, basic 1st, basic cycle 2nd, basic cycle 3rd, secondary cycle, upper middle), employment status, inactive and active population per 100, level of purchasing power in 2019, total population, dependence index of elderly from the 2021 census, aging index from the 2021 census, and longevity index from the 2021 census.

### 2.3. Sample Size

The sample size used for periodontal prevalence estimates was derived from secondary data previously collected in the SoPHiAS study. The sample size has further explained elsewhere [12]. This study was approved by the Research Ethics Committee of the Regional Health Administration of Lisbon and Tagus Valley, IP (Portugal) (approval 3525/CES/2018 and 8696/CES/2018) and each participant provided signed informed consent prior to evaluation.

Originally, we based the sample-size estimation on the reported national prevalence data of 10.8% and 15.3% for adults and elderly, respectively [11]. For a minimum sample of 962 individuals, we were able to include 1064 participants, stratified according to the number of adults (age-group 18–64 years) and elderly (65 years or older) subjects assigned to each health center of the Almada-Seixal health centers group. The periodontal diagnosis followed the American Academy of Periodontology (AAP)/European Federation of Periodontology (EFP) consensus [3] based on a circumferential recording (six sites per tooth—mesiobuccal, buccal, distobuccal, mesiolingual, lingual, and distolingual).

### 2.4. Statistical Analysis

Principal component analysis (PCA), factor analysis (FA), and cluster analysis (CA) were used to model the nationwide geographical distribution of the disease. Through the PCA/FA dimension reduction process, six factors were extracted from the original set of 52 variables. The factors were further used to categorize the municipalities through a CA procedure.

Additionally, linear regression was used to estimate the periodontitis prevalence within the peri-urban municipality clusters (18 municipalities), accounting for 30.5% of the total Portuguese population. The estimations were based on the calculated risk score and prevalence values previously obtained in the SoPHiAS, which was conducted in two peri-urban municipalities in the Lisbon Metropolitan Area [12]. All analyses were conducted through IBM SPSS Statistics 28.

Using an online map-creator website (mapchart.net) we created a customized map for the risk of periodontitis on the Portuguese mainland and autonomous islands per region. To create this map, we categorized the risk per quartile as described in the Section 2.2.

## 3. Results

### 3.1. Model Development

Prior to the model development and to reduce the complexity of the dimension data, the 52 variables were grouped into six factors through a PCA/FA extraction procedure. The distribution of variables among the factors was as follows: factor 2—inactive and active population for female and male subjects, dependence index of elderly, and longevity index; factor 3—number of doctors and nurses, and level of purchasing power; factor 4—employment status for female and male subjects; factor 5—illiterates in total and male/female ratio; factor 6—unemployment ratio; factor 1—all variables except those mentioned in the remaining factors. The factors were used to categorize the municipalities in clusters.

The developed model resulted in a 9-cluster based distribution (Table 1). In this, 4 clusters were based on one municipality each (Porto, V. N. Gaia, Lisbon and Sintra), two were based on eight municipalities each (clusters 4 and 5) and two clusters were based on over 100 municipalities (clusters 1–3). We also obtained a periodontitis normalized risk score and range for each cluster (Table 2) that contributed to the development of the model for the risk towards periodontitis. In this, Lisbon (25.11) and Porto (23.52) presented the higher normalized risk scores while municipalities of cluster 1 had the lowest (9.69).

### 3.2. Periodontitis Risk and Prevalence Estimation

When analyzing the map encompassing the predicted risk score for each cluster, the major population areas in Portugal, particularly the Lisbon and Porto metropolitan areas, have the largest population high-risk clusters for periodontitis (Figure 1). Furthermore, the less populated areas seem to at lower risk, areas populated by older people, with higher levels of tooth loss.

When forecasting the estimated periodontitis prevalence, the prevalence ranged between 41.4% and 69.0% (Table 3). The lowest estimated prevalence was in Leiria, while Funchal presented the highest score.

## 4. Discussion

The present study was able to predict estimates for risk of periodontitis on the Portuguese mainland and autonomous regions (Azores and Madeira) based on sociodemographic/economic reference data from national governmental sources. Additionally, it was possible to produce estimates for periodontitis prevalence from secondary data previously obtained in a regional field study. The periodontitis prevalence estimates range from 41.2% to 69.0%, for the municipalities characterized as peri-urban and mainly urban, including some medium-sized to large cities. Most of them are located in the Lisbon Metropolitan Area (NUTS 2, PT170), which is the largest urban area in the country and the tenth largest in Europe. However, it is worth noting that the highest prevalence (69.0%) was estimated for Funchal municipality, in the Madeira autonomous region. These findings contrast with those presented in the DGS oral health study in 2015. In other words, and according to DGS data, the prevalence of periodontitis in Madeira was 6.6% and 10.8% in those aged 35–44 and 65–74 years, respectively. Nevertheless, our results are similar for other regions, and ultimately confirm previous epidemiological studies conducted mainly in the Portuguese metropolitan areas (Lisbon and Porto) [12,23,24]. In particular, the estimates for the Porto Metropolitan Area were slightly above those estimated by Relvas et al. [23], yet this study was conducted in a university dental clinic and this could explain the slight differences due to lower generalizability of the results by this prevalence study. Nonetheless, the prevalence estimates obtained and the agreement with previous studies corroborate alarming levels of periodontitis in Portugal and the demand for periodontal health programs along with the Health General Directorate and health center groupings.

A lack of financial support has often been a serious obstacle, in most countries, to permit adequate and sufficient dental care [25]. In Portugal, most oral health services are provided by private dentists who are not associated or have any agreement with public health protection mechanisms. Thus, in cases of illness, it is often the patients who pay the full cost of treatment, which often creates problems in accessing oral health care [25].

In 2008, a “dental voucher” strategy was added to the program of the Portuguese National Health Service (NHS). Dental vouchers are awarded by primary care health centers to certain beneficiary patients, such as children, pregnant women followed in the NHS, receivers of specific social protection benefits, and people diagnosed with human immunodeficiency virus. This allows access to a range of preventive and curative treatments, free of charge, at any private dental surgery that has an agreement with the NHS [25]. Finally, another strategy was defined in 2016, which consisted of the integration of dentists in the performance of primary oral health care in health centers in order to provide treatment to patients referred by family doctors [25].

Despite the measures adopted, periodontitis remains one of the diseases that most affects the oral cavity in Portugal [12]. It is therefore so important to have studies that show us the prevalence of this disease so that measures can be adopted to counter this trend [26]. Epidemiological data can form the basis for selection and implementation of strategies to prevent and treat periodontal diseases. Three broad strategies have been advanced: a community-wide approach in which health education and other favorable life practices are introduced into the community; a secondary prevention strategy which includes detecting and treating individuals with destructive periodontal diseases; and identification of groups at high risk of periodontitis [27,28].

Given this, the present study acquires extreme importance. It is an innovative study that has never been carried out before and that gives us information about periodontitis risk by county. Even so, this study presents some limitations due to the lack of information on sociodemographic and socioeconomic issues and the lack of information on oral health habits, such as tooth brushing per day and interproximal cleaning. In this way, this study was carried out by the county and not by the parish, as it had been initially planned. Analyzing the risk map, within each cluster, it is possible to identify the counties with the greatest risk for periodontitis based on a score from 1 to 4, where 4 represents the greatest risk.

The values obtained in this study show a high prevalence of periodontitis in Portugal, the second most common disease concerning the oral cavity. Thus, the need for greater inclusion of dentistry in health centers is foreseen, with a greater interconnection between dentistry and the national health service, and a greater coverage of the treatments available.

There are limitations worth mentioning. The results of these forecasts are always dependent on the availability, amount, and quality of data used to model the results. Although the data available could assemble a model, the amount is far from ideal, and requires further enhancement in the future. No prediction is entirely exact and possible variations or deviations from these forecasts are mandatory. Nevertheless, the periodontal data used were based on updated periodontal case definitions and high-quality epidemiological designs, hitherto confined to the Lisbon Metropolitan Area; therefore, national and regional studies should be carried out on a regular basis to allow consistent results and forecasting. Despite the abovementioned limitations and strengths, we developed a model to identify risk per municipality of periodontitis in Portugal. This geographical distribution of periodontitis risk allows region targeting for adequate planning of oral health policy. These findings provide important data to explore through regional/territory-level epidemiological studies.

## 5. Conclusions

The periodontitis prevalence estimates range from 41.2% to 69.0% for the municipalities characterized as peri-urban and mainly urban, most of them located in the Lisbon Metropolitan Area, the tenth largest in Europe.

## Figures and Tables

**Figure 1 ijerph-19-13634-f001:**
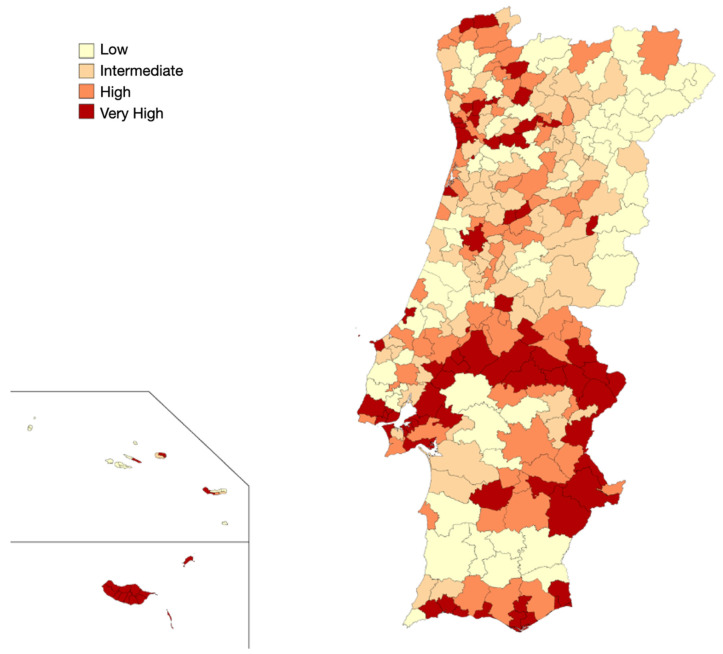
National distribution of the periodontitis risk grade for each LAU 1 (municipality) within the correspondent cluster. The municipalities are identified in a 4-level color risk grade, by quartile position: from Q1—higher risk (red) to Q4—lower risk (yellow).

**Table 1 ijerph-19-13634-t001:** Local Administrative Units 1 (municipalities) cluster distribution. Table reports the sociodemographic and number of municipalities.

Cluster	Sociodemographic Characteristics	N	Municipalities
1	mainly rural/low populated, including small villages	143	(*)
2	partly rural/including small cities	104	(*)
3	partly rural/including small to medium cities	41	(*)
4	mainly urban/peri-urban/including medium-sized to large cities	8	Leiria, Barcelos, S. Maria Feira, V. F. Xira, V. N. Famalicão, Maia, Setúbal, Funchal
5	mainly urban/peri-urban/including medium-sized to large cities	8	Guimarães, Loures, Braga, Seixal, Gondomar, Cascais, Almada, Matosinhos
6	urban/large city	1	Porto
7	peri-urban/large city	1	V. N. Gaia
8	urban/large city	1	Lisbon
9	peri-urban/large city	1	Sintra

(*) full list in Appendix A.

**Table 2 ijerph-19-13634-t002:** Periodontitis normalized risk score for each cluster.

Cluster	N	Average	Range (Min.–Max.)	Observations
1	143	9.69	0.00–15.76	-
2	104	11.11	3.72–16.51	-
3	41	12.27	9.40–16.66	-
4	8	12.57	10.84–15.72	-
5	8	13.69	12.13–15.64	-
6	1	23.52	-	Only one municipality (Porto)
7	1	13.96	-	Only one municipality (V. N. Gaia)
8	1	25.11	-	Only one municipality (Lisbon)
9	1	10.81	-	Only one municipality (Sintra)

**Table 3 ijerph-19-13634-t003:** Estimation of periodontitis prevalence (%) for the municipalities integrating the peri-urban clusters with reference to the geographical distribution under Eurostat Nomenclature of Territorial Units for Statistics (NUTS) region codes (NUTS 1 and 2).

Cluster	Municipalities	Population (n)	NUTS 1	NUTS 2	Estimated Periodontitis Prevalence (%)
4	Leiria	128,640	PT16	PT16F	41.4
	Barcelos	116,777	PT11	PT112	41.6
	S. Maria Feira	136,720	PT11	PT11A	46.7
	V. F. Xira	164,255	PT17	PT170	47.5
	V. N. Famalicão	134,959	PT11	PT112	48.2
	Maia	134,959	PT11	PT11A	57.1
	Setúbal	128,640	PT17	PT170	57.7
	Funchal	105,919	PT30	PT300	69.0
5	Guimarães	164,255	PT11	PT119	48.7
	Loures	201,646	PT17	PT170	53.3
	Braga	193,333	PT11	PT112	54.9
	Seixal	166,693	PT17	PT170	55.2 (*)
	Gondomar	164,255	PT11	PT11A	55.5
	Cascais	214,134	PT17	PT170	60.9
	Almada	177,400	PT17	PT170	63.3 (*)
	Matosinhos	172,669	PT11	PT11A	68.6
7	V. N. Gaia	304,149	PT11	PT11A	59.1
9	Sintra	385,954	PT17	PT170	41.2

(*) Values previously obtained from the field study SoPHiAS [12]. Abbreviations: n—number of subjects; NUTS—nomenclature of territorial units for statistics; PT—Portugal.

## Data Availability

Data will be available upon request from the corresponding author.

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
