# Peer review of "Geographical Distribution of Periodontitis Risk and Prevalence in Portugal Using Multivariable Data Mining and Modeling"

_ijerph, 2022, doi:10.3390/ijerph192013634_

Round 1

Reviewer 1 Report

To authors:

I congratulate and encourage the authors to finish this interesting manuscript with the requested suggestions to be corrected in the text.

The study methodology is carefully designed and architected. The analysis of the data through the statistical tests described presents a scientific robustness.

The discussion is linked to the results observed in this and other studies and the objectives converge with their conclusions.

2. Introduction

The introduction is well updated and well linked to the theme and basic objectives of this study.

Corrections: pg. 2 line 50, there is a reference to REFs at the end of the text. Please insert the established references for the context you are referring to.

3. Materials and methods

Methodologically, the study and the form of data collection are interesting, however the Ethical aspects were considered?? Legal aspects of patient consent is not clear in the methodology and should at least be mentioned.

Results and discussion

The inferences regarding results obtained are quite clear.

The study clearly presents the strengths/limitations and implications of findings for research and practice.

Would it be possible to extrapolate the results presented here and compare them with similar studies from other European or non-European countries?

Conclusions:

Clear, objective and in accordance with the proposed objectives and results obtained.

I congratulate the authors for the excellent material presented in this research and for the scientific maturity described in this research.

Author Response

Editors of 
International Journal of Environmental Research and Public Health

We are delighted to have the opportunity to revise and resubmit our manuscript titled “Geographical distribution of periodontitis risk and prevalence in Portugal using multivariable data mining and modeling” (Manuscript ID ijerph-1968075).
We have considered the editorial and reviewers’ comments. Please find appended a track-changes draft of the manuscript and a point-by-point rebuttal to all comments raised as detailed below. We hope our responses are satisfactory in addressing the criticisms and suggestions.
We hope the revised manuscript will be in acceptable format for your journal.

Referee: 1
To authors:
I congratulate and encourage the authors to finish this interesting manuscript with the requested suggestions to be corrected in the text.
The study methodology is carefully designed and architected. The analysis of the data through the statistical tests described presents a scientific robustness.
The discussion is linked to the results observed in this and other studies and the objectives converge with their conclusions.
Our answer: We appreciate this remark very much. 

2. Introduction
The introduction is well updated and well linked to the theme and basic objectives of this study.
Corrections: pg. 2 line 50, there is a reference to REFs at the end of the text. Please insert the established references for the context you are referring to.
Our answer: We apologize for this informatic lapse. We have provided the reference accordingly.

3. Materials and methods
Methodologically, the study and the form of data collection are interesting, however the Ethical aspects were considered?? Legal aspects of patient consent is not clear in the methodology and should at least be mentioned.
Our answer: We appreciate this observation, and initially we did not report the ethical aspects because we explained that the SoPHiAS database was “further explained elsewhere” (Botelho et al. 2019). Nevertheless, and following this valid recommendation we provided this information by adding: “This study was approved by the Research Ethics Committee of the Regional Health Administration of Lisbon and Tagus Valley, IP (Portugal) (Approval numbers: 3525/CES/2018 and 8696/CES/2018) and each participant provided signed informed consent prior to evaluation.” (Page 3, Line 95-98).
Botelho, J.; Machado, V.; Proença, L.; Alves, R.; Cavacas, M.A.; Amaro, L.; Mendes, J.J. Study of Periodontal Health in Al-mada-Seixal (SoPHiAS): A Cross-Sectional Study in the Lisbon Metropolitan Area. Sci Rep 2019, 9, 15538, doi:10.1038/s41598-019-52116-6

Results and discussion
The inferences regarding results obtained are quite clear.
The study clearly presents the strengths/limitations and implications of findings for research and practice.
Our answer: We appreciate acknowledging this.

Would it be possible to extrapolate the results presented here and compare them with similar studies from other European or non-European countries?
Our answer: We appreciate this remark very much, and we intend to explore this in the future. To do so, we will be required to have access to paralleled information from other countries from European and national databases and ensure the predictive factors are comparable to all countries. Currently, this is not possible due to the variability of data, but we believe this study may contribute to realize this need and increase the number of standardized databases.

Conclusions:
Clear, objective and in accordance with the proposed objectives and results obtained.
Our answer: We appreciate acknowledging this.

I congratulate the authors for the excellent material presented in this research and for the scientific maturity described in this research.
Our answer: We are thankful for these encouraging words and your time and effort invested in reviewing our manuscript.

Reviewer 2 Report

This manuscript identifies the periodontitis prevalence and risk in Portugal using sociodemographic, economic and health services data with a set of multivariate analysis techniques. Generally, the manuscript wrote plainly. The analysis, presentation, and discussion can be improved by focusing on “periodontitis prevalence and risk prediction”. Why sociodemographic and economic data can be used to predict the periodontitis prevalence and risk? This can be mentioned in Introduction or discussed in the discussion. Furthermore, I can not see the supplementary files. These issues can be addressed to make the manuscript more impressive.

L 50: add refs

Materials and Methods

L 61-109:  More details should be added to make the methods, data, analytical technique clearer. This section has three “Statistical Analysis”. These repeated subtitles can be replaced by more precise descriptions, for example, Study area/data collection, multivariate analysis.

L70-72: How is the normalized score estimated? Why it ranges from 0-25? How are the municipality risk grade (Q1-Q4) identified. Detailed description is necessary. To clarify, you can use formula here.

Results

Why do not present the PCA, FA, and CA plots?  Coloured by regions or municipalities?

What are the relations between “sociodemographic, economic and health services” and periodontitis prevalence and risk?

If you model periodontitis prevalence and risk using sociodemographic, economic and health services, there should be some model performance metrics to show how reliable it is.

Fig.1.

The legend should be in English.

Discussion

Why periodontitis prevalence and risk are high in some areas can be discussed.

The manuscript may show us that sociodemographic and economic data can be used to predict the periodontitis prevalence and risk, rather than emphasizing the geographical distribution. Details about how the authors modelled the periodontitis prevalence and risk, and what is the model performance, as well as what periodontitis prevalence and risk (high/ medium /low) reflected by sociodemographic and economic data and why this could happen should be described.  

Author Response

Editors of 
International Journal of Environmental Research and Public Health

We are delighted to have the opportunity to revise and resubmit our manuscript titled “Geographical distribution of periodontitis risk and prevalence in Portugal using multivariable data mining and modeling” (Manuscript ID ijerph-1968075).
We have considered the editorial and reviewers’ comments. Please find appended a track-changes draft of the manuscript and a point-by-point rebuttal to all comments raised as detailed below. We hope our responses are satisfactory in addressing the criticisms and suggestions.
We hope the revised manuscript will be in acceptable format for your journal.

Referee: 2
This manuscript identifies the periodontitis prevalence and risk in Portugal using sociodemographic, economic and health services data with a set of multivariate analysis techniques. Generally, the manuscript wrote plainly. The analysis, presentation, and discussion can be improved by focusing on “periodontitis prevalence and risk prediction”. Why sociodemographic and economic data can be used to predict the periodontitis prevalence and risk? This can be mentioned in Introduction or discussed in the discussion. Furthermore, I can not see the supplementary files. These issues can be addressed to make the manuscript more impressive.
Our answer: Thanks for pointing this out. In fact, these data can be used to calculate the risk of periodontitis as they consist of risk factors proven statistically relevant in a previous regional study (SoPHiAS). We emphasize that this is the only epidemiologic study of this kind using gold-standard full-mouth periodontal examination and the more recent periodontitis case definition. In this, the risk of periodontitis significantly increased with age (OR = 1.05, 95% CI: 1.04–1.06), for active and former smokers (OR = 3.76 and OR = 2.11, respectively), with lower education levels (OR = 2.08, OR = 1.86, for middle and elementary education, respectively) and with diabetes mellitus (OR = 1.53). Subsequent studies further explored this in adult and elderly subsets, with the economic determinant being relevant in this “equation” (Machado et al. 2020, Botelho et al. 2020). Therefore, based on this, sociodemographic and economic data were used as predictors to model the periodontitis prevalence and risk.
Regarding the supplementary file, by informatic lapse this was not uploaded, yet we resolved this issue and now the supplementary file is available.
_________________
References
Machado, V., Botelho, J., Proença, L. et al. Periodontal status, perceived stress, diabetes mellitus and oral hygiene care on quality of life: a structural equation modelling analysis. BMC Oral Health 20, 229 (2020). https://doi.org/10.1186/s12903-020-01219-y
Botelho, J., Machado, V., Proença, L. et al. Perceived xerostomia, stress and periodontal status impact on elderly oral health-related quality of life: findings from a cross-sectional survey. BMC Oral Health 20, 199 (2020). https://doi.org/10.1186/s12903-020-01183-7

L 50: add refs
Our answer: This was a typo, we appreciate pointing this out. We have provided the reference information.

Materials and Methods
L 61-109:  More details should be added to make the methods, data, analytical technique clearer. This section has three “Statistical Analysis”. These repeated subtitles can be replaced by more precise descriptions, for example, Study area/data collection, multivariate analysis.
Our answer: Following your suggestion, we merged subsections 2.1 and 2.2 into a new common subsection “2.1. Study area”.

L70-72: How is the normalized score estimated? Why it ranges from 0-25? How are the municipality risk grade (Q1-Q4) identified. Detailed description is necessary. To clarify, you can use formula here.
Our answer: The score was calculated according to a linear combination of the 6-factor loading weights for each municipality. For example, considering the municipality of Almada, its 6-factor loadings were: F1 = 2.4948; F2 = 0.09643; F3= -0,20074; F4 = 0.36004; F5 = 0.13562; F6 = 1.01811.
The formula for calculating the overall score was based on an algebraic sum of the factor loadings: F1+F2+F3+F4+F5+F6. It is relevant to note that factor loadings were obtained by regression through the PCA/factor analysis procedure. Thus, based on the formula, the non-normalized score for the Almada municipality was: 3.90.
In a second stage the score was normalized across clusters, taking into account the overall minimum score obtained, allowing direct comparison between municipalities among different clusters. For the referred example, the score was normalized into 14.71. The obtained normalized score for all 308 municipalities ranged between 0 and 25. The numerical range (0-25), per se, is not relevant, since it was a consequence of the normalization process. The importance here is the ability to compare municipalities intra and across clusters.
Risk grade was based on a 4-quartile grade index for classification of the individual scores. This was performed as an intra-cluster risk grade assessment. Each municipality was categorized according to its position on 4-quartile division, from Q1 (higher quartile - higher risk) - to Q4 (lower quartile - lower risk):  Q1 - 75-100%, Q2 - 50-75%, Q3 - 25-50% and Q4 - 0-25%

Results
Why do not present the PCA, FA, and CA plots?  Coloured by regions or municipalities?
What are the relations between “sociodemographic, economic and health services” and periodontitis prevalence and risk?
Our answer: We chose to not present the PCA, FA and CA plots due to their fairly complexity to be interpreted as you may see in the example below. We hope this reviewer can understand this fact deterred us from placing it in the manuscript.

If you model periodontitis prevalence and risk using sociodemographic, economic and health services, there should be some model performance metrics to show how reliable it is.
Our answer: We considered this comment as extremely valid. Having this said, we want to reinforce the exploratory design of the present study. In the near future, we plan to validate this model through an epidemiological study in some municipalities with different degrees of risk of periodontitis, according to these results. Furthermore, since the validation of the model requires epidemiological data on periodontitis by municipalities, it is currently not possible for us to perform such a type of statistical analysis. However, we will take this into consideration for the upcoming epidemiological studies that are being planned with regional and national entities.

Fig.1. The legend should be in English.
Our answer: We apologize for this informatic lapse. We have provided the correct figure. 

Discussion
Why periodontitis prevalence and risk are high in some areas can be discussed.
Our answer: We consider this commentary as extremely valid, but after considering it with all the team members, it seems that trying to justify the risk by the characteristics of each municipality could lead to biased and wrong interpretations. Therefore, we chose not to add this information and we emphasize the pilot nature of this study that points to the possible municipalities risk towards periodontitis and the ultimate need of confirmation of the results through a national epidemiological study. By discussing abroad we could contribute to diluting the importance of unknown results to be discussed with further studies.

The manuscript may show us that sociodemographic and economic data can be used to predict the periodontitis prevalence and risk, rather than emphasizing the geographical distribution. Details about how the authors modelled the periodontitis prevalence and risk, and what is the model performance, as well as what periodontitis prevalence and risk (high/ medium /low) reflected by sociodemographic and economic data and why this could happen should be described.  
Our answer: We appreciate this commentary. Indeed we have described such information on the Statistical analyses, specifically in paragraph 2 (lines 118-121). We hope this may have clarified this remark.

Reviewer 3 Report

Abstract:

1. the first 2 sentences are redundant. It can be rephrase into one sentence to give meaningful info in the abstract

Introduction:

1. The authors could highlight the importance of this study rather than stating 'a few studies has been conducted' as in line 41

2. sentence from line 47 must have references

Material and methods:

1. there were multiples subheadings for 'Statistical Analysis', which 2 of them does not necessary be stand for subheadings. subheading 2.2 and 2.2 can be renamed, or incorporate into existing heading.

Results: 

1. Figure 1 does not make sense. It was not appropriately presented to make the results outstanding. The label was in Portugese (Baixo, Intermedio, Alto, Muiytop Alto - were not English terms).

2. it didnot mentioned why only 18 municipalities included in the result for prevalence, is it because those 18 are the considered as peri-urban, or these 18 were the top quarter of higher risk

conclusions:

could suggest why the conclusion is as such by adding one sentence following existing conclusion.

Author Response

Editors of 

We are delighted to have the opportunity to revise and resubmit our manuscript titled “Geographical distribution of periodontitis risk and prevalence in Portugal using multivariable data mining and modeling” (Manuscript ID ijerph-1968075).

We have considered the editorial and reviewers’ comments. Please find appended a track-changes draft of the manuscript and a point-by-point rebuttal to all comments raised as detailed below. We hope our responses are satisfactory in addressing the criticisms and suggestions.

We hope the revised manuscript will be in acceptable format for your journal.

Referee: 3

Abstract:

  1. The first 2 sentences are redundant. It can be rephrase into one sentence to give meaningful info in the abstract

Our answer: We respectfully disagree with the reviewer commentary. In the first phase we refer to what is periodontitis (a disease that affects the tissues around the teeth) and the main etiological factor (polymicrobial dysbiosis). In addition, in the second sentence, we explain the personal and economic impact of periodontitis. In our point-of-view, both information are fundamental for the readers of IJERPH, which is not a specific journal of the periodontology field, to understand the etiology and consequences of periodontitis in the population. However, we are available to rewrite if the editor deems it necessary.

Introduction:

  1. The authors could highlight the importance of this study rather than stating 'a few studies has been conducted' as in line 41

Our answer: We have revised this sentence following the reviewer recommendation by re-phrase to “To date, the epidemiology of periodontal diseases in the Portuguese population was inadequately reported.” (Page 1, Line 41-42)

  1. sentence from line 47 must have references

Our answer: This was a typo, we appreciate pointing this out. We have provided the reference information.

Material and methods:

  1. there were multiples subheadings for 'Statistical Analysis', which 2 of them does not necessary be stand for subheadings. subheading 2.2 and 2.2 can be renamed, or incorporate into existing heading.

Our answer: Following your suggestion, we merged subsections 2.1 and 2.2 into a new common subsection “2.1. Study area”.

Results: 

  1. Figure 1 does not make sense. It was not appropriately presented to make the results outstanding. The label was in Portugese (Baixo, Intermedio, Alto, Muiytop Alto - were not English terms).

Our answer: We apologize for this informatic lapse. We have provided the correct figure. 

  1. it didnot mentioned why only 18 municipalities included in the result for prevalence, is it because those 18 are the considered as peri-urban, or these 18 were the top quarter of higher risk

Our answer: We are thankful for this question. Taking this opportunity to clarify this point, the prevalence was estimated based on real data from the Almada and Seixal regional study, for municipalities with similar peri-urban characteristics (the only epidemiological data available at the time). Hence, it only makes sense to extrapolate values to other municipalities with similar socio-economic characteristics (supported by the fact that they are part of the same cluster where Almada and Seixal - Cluster 5 are found, or, if they are not, the case of V.N. Gaia and Sintra, and the municipalities in cluster 4, with scores similar to Almada/Seixal). We hope this clarification was sufficient to this reviewer, and we are available for further explanations if you find it necessary.

conclusions: could suggest why the conclusion is as such by adding one sentence following existing conclusion.

Our answer: We respectfully apologize, yet we could not understand your remark. However, we underline that conclusion was based on IJERPH previous studies that followed their manuscript template.

Round 2

Reviewer 2 Report

The authors seemed reply some of the questions, but the presentation, interpretation of the analysis still need improvement. If this manuscript just describes the geographic distribution of the periodontitis risk and prevalence, I do not have any other comments, but apparently the novelty can be added and improved by addressing different points as I raised in the first round review.

Author Response

We are delighted to have the opportunity to revise  and resubmit our manuscript titled “Geographical distribution of periodontitis risk and prevalence in Portugal using multivariable data mining and modeling” (Manuscript ID ijerph-1968075).

We have considered the reviewer comment. Please find appended a track-changes draft of the manuscript and a point-by-point rebuttal to all comments raised as detailed below. We hope our responses are satisfactory in addressing the criticisms and suggestions. We hope the revised manuscript will be in acceptable format for your journal.

The authors seemed reply some of the questions, but the presentation, interpretation of the analysis still need improvement. If this manuscript just describes the geographic distribution of the periodontitis risk and prevalence, I do not have any other comments, but apparently the novelty can be added and improved by addressing different points as I raised in the first round review.

Our answer: We are thankful for this question. Following your suggestion, we have provided more information regarding the variables included in each factor in the results section, by adding: “Prior to the model development, and to reduce the complexity of the dimension data, the 52 variables were grouped into six factors, through linear regression. The distribution of variables in the factors was as follows: factor 2 – inactive and active population for female and male subjects, dependence index of elderly, and longevity index; factor 3 - number of doctors and nurses, and level of purchasing power; factor 4 - employment status for female and male subjects; factor 5 - illiterates in total and male/female ratio; factor 6 - unemployment ratio; factor 1 - all variables except those mentioned in the remaining factors. The factors were used to categorize the municipalities in clusters.” (Page 3, Lines 131-138)

Also, we also added information on the discussion section to emphasize the main results and to compare with other studies, by adding these sentences: 

  • "These findings contrast with those presented in the DGS oral health study in 2015. In other words, and according to DGS data, the prevalence of periodontitis in Madeira was 6.6% and 10.8% in 35-44 and 65-74 years of age, respectively. Nevertheless, our are similar in other regions, and ultimately attest previous epidemiological studies conducted mainly in the Portuguese metropolitan areas (Lisbon and Porto)” (Page 6, Line 192-196)
  • “Even so, this study presents some limitations due to the lack of information on the so-ciodemographic and socioeconomic issues, and, also, the lack of oral health habits such as tooth brushing per day and interproximal cleaning.” (Page 6, line 228-230)
  • “Although the abovementioned limitations and strengths, we developed a model to identify the risk per municipalities towards periodontitis in Portugal. This geographical distribution of periodontitis risk allows each region-targeted to adequate planning of the oral health policy. Furthermore, in the future, these findings provide important data to explore through regional/territory-level epidemiological studies. “ (page 7, lines 225-260)